# Building a Human Physiologically Based Pharmacokinetic Model for Aflatoxin B1 to Simulate Interactions with Drugs

**DOI:** 10.3390/pharmaceutics15030894

**Published:** 2023-03-09

**Authors:** Orphélie Lootens, Marthe De Boevre, Jia Ning, Elke Gasthuys, Jan Van Bocxlaer, Sarah De Saeger, An Vermeulen

**Affiliations:** 1Centre of Excellence in Mycotoxicology and Public Health, Department of Bioanalysis, Ghent University, 9000 Ghent, Belgium; 2Laboratory of Medical Biochemistry and Clinical Analysis, Department of Bioanalysis, Ghent University, 9000 Ghent, Belgium; 3MYTOX-SOUTH®, International Thematic Network, 9000 Ghent, Belgium; 4Cancer Research Institute Ghent (CRIG), 9000 Ghent, Belgium; 5Simcyp Division, Certara UK Limited, Sheffield S1 2BJ, UK; 6Department of Biotechnology and Food Technology, University of Johannesburg, Gauteng 2028, South Africa

**Keywords:** PBPK, aflatoxin B1, mycotoxins, IVIVE, DDI, food contaminants

## Abstract

Mycotoxins such as aflatoxin B1 (AFB1) are secondary fungal metabolites present in food commodities and part of one’s daily exposure, especially in certain regions, e.g., sub-Saharan Africa. AFB1 is mostly metabolised by cytochrome P450 (CYP) enzymes, namely, CYP1A2 and CYP3A4. As a consequence of chronic exposure, it is interesting to check for interactions with drugs taken concomitantly. A physiologically based pharmacokinetic (PBPK) model was developed based on the literature and in-house-generated in vitro data to characterise the pharmacokinetics (PK) of AFB1. The substrate file was used in different populations (Chinese, North European Caucasian and Black South African), provided by SimCYP^®^ software (v21), to evaluate the impact of populations on AFB1 PK. The model’s performance was verified against published human in vivo PK parameters, with AUC ratios and C_max_ ratios being within the 0.5–2.0-fold range. Effects on AFB1 PK were observed with commonly prescribed drugs in South Africa, leading to clearance ratios of 0.54 to 4.13. The simulations revealed that CYP3A4/CYP1A2 inducer/inhibitor drugs might have an impact on AFB1 metabolism, altering exposure to carcinogenic metabolites. AFB1 did not have effects on the PK of drugs at representative exposure concentrations. Therefore, chronic AFB1 exposure is unlikely to impact the PK of drugs taken concomitantly.

## 1. Introduction

Mycotoxins are secondary metabolites produced by fungi that are frequently present in food commodities. Mycotoxins cause major health problems in both humans and animals [1]. Aflatoxin B1 (AFB1) is a human carcinogenic, hepatotoxic and immunotoxic mycotoxin, produced by *Aspergillus flavus* and *Aspergillus parasiticus,* and it is classified as Group 1 by the International Agency for Research on Cancer (IARC) [2,3,4,5]. Figure 1 represents the metabolic pathway of AFB1 in the human liver. AFB1 is metabolised mainly by CYP1A2 and CYP3A4 [6,7,8,9]. CYP1A2 facilitates the formation of both aflatoxin M1 (AFM1), classified as Group 2B by IARC as potentially carcinogenic to humans, and aflatoxin-endo/exo-8,9-epoxide (AFBO), carcinogenic metabolites with aflatoxin-exo-8,9-epoxide being the more potent carcinogenic metabolite [2,6,7,8,9]. The metabolite AFBO is carcinogenic since the reactive epoxide can form adducts with DNA [6,10]. Aflatoxin P1 (AFP1), considered less toxic than AFB1, is formed by CYP2A3, CYP2A13 and CYP321A1 [7,11]. CYP3A4 facilitates the formation of aflatoxin Q1 (AFQ1), a detoxification product with less toxicity than AFB1, and carcinogenic AFBO metabolites [12].

Nicotinamide-adenine dinucleotide phosphate (NADPH) reductase is responsible for the formation of aflatoxicol (AFL), a detoxification product of AFB1 [11,13]. Both AFQ1 and AFL can be further metabolised into AFH1 [14]. Aflatoxins (AFs) are found in cereals, spices, soybeans, crude vegetable oils, seeds, rice, dried fruits, maize and nuts [15]. In certain regions of the world, mycotoxin prevalence is endemic, e.g., sub-Saharan Africa. A humid and warm climate promotes fungal growth [16]. Stress situations (e.g., drought, rises in temperature or humidity) further cause the fungi to produce mycotoxins [17]. Besides a higher mycotoxin prevalence on the African continent, there is also a lack of mycotoxin regulation legislation and/or implementation thereof [18]. This is partly related to food shortages, cultural habits, informal markets and subsistence farming. Another major reason for higher mycotoxin exposure is the lack of mycotoxin awareness. This results from both a lack of public awareness and a lack of awareness from the actors in the food and feed chain [19]. In 2004, 2005 and 2006, mycotoxin outbreaks caused by AFs led to hundreds of deaths in Kenya [20,21]. In Tanzania, both children and elderly people died after consuming AF-contaminated homegrown foods in 2016 [22]. In 2019, 8 out of 53 cases died from a mycotoxin outbreak, again due to AFs [23]. Repeated exposure to mycotoxins not only leads to severe health effects but might also lead to interactions with drugs, influencing plasma concentration and altering mycotoxin exposure. Because of the carcinogenicity of AFB1, human in vivo trials are not considered ethical. A solution is to perform in silico simulations of the mycotoxin–drug interactions using physiologically based pharmacokinetic (PBPK) modelling, allowing us to predict the outcomes of concomitant intake without having to perform clinical trials.

PBPK models describe the body using physiologically meaningful compartments that are interconnected through the circulating blood system. Within each compartment, drug-dependent, body-dependent and trial-dependent factors are incorporated dynamically. PBPK modelling is fully embedded in the pharmaceutical industry across the different drug discovery and preclinical/clinical development stages. When mechanistic PBPK models are validated against clinical conditions, they can be used to extrapolate or to predict certain situations beyond the clinical studied scenarios. One of the strengths of PBPK is drug–drug interaction (DDI) predictions that are frequently used to guide dosing recommendations to manage DDIs. Additionally, some intrinsic and extrinsic factors, such as disease state, ethnicity, and age-related factors, which can be challenging to address in clinical studies due to ethical or practical reasons, can be taken into account. Since PBPK models are commonly used to predict DDIs clinically, they could also play a role in the prediction of food contaminant–drug interactions. Currently, the presence or absence of food (fasted/fed status) is considered in clinical trials and simulations, but the food components are generally not taken into account. Chronic food contaminants may have an impact on the PK or PD of drugs and should be considered. However, in contrast to drugs, these contaminants are often present in much lower quantities, making it likely that drugs might have an impact on the PK of the food contaminants rather than vice versa. PBPK models and physiologically based toxicokinetic (PBTK) models have previously been built for mycotoxins, such as for the T-2 toxin (T-2) in a chicken PBPK model [24], and deoxynivalenol (DON) in a human PBPK model [25]. A PBK model was developed for AFB1 based on in vitro–in silico by Gilbert-Sandoval et al., 2020 [26]. The mentioned models were developed to simulate exposure to a certain mycotoxin. In this paper, a PBPK substrate file of AFB1 is developed using published physicochemical data and in-house-generated enzyme kinetic data and is verified against in vivo PK in a Chinese population. Subsequently, the PK differences in North European Caucasian, Chinese and Black South African populations are assessed. Finally, the interactions between AFB1 and commonly prescribed drugs in South Africa are evaluated.

## 2. Materials and Methods

The PBPK model of AFB1 was developed in the SimCYP^®^ population-based simulator (SimCYP Ltd., a Certara company, Sheffield, UK, version 21). A stepwise bottom-up approach was applied while building the PBPK model and is represented in a flowchart (Figure 2). The input parameters and references are shown in Table 1. Physicochemical properties, absorption, metabolism and excretion parameters and interaction and transport data were retrieved from the literature and from computations. The distribution in the PBPK model was represented by the Rodgers et al. model [27] and by incorporating sub-cellular distribution (Method 3 in the SimCYP^®^ simulator version 21). The final PBPK model was evaluated by comparison with human in vivo PK data [28].

SimCYP^®^ has an available pilot Black South African population file (*SouthAfrican_Population FW_Custom*), later named *Black South African population*, that was built from an already available North European Caucasian population (*Sim-NEur Caucasian population*).

For the age distribution, a Weibull function was selected. For both males and females, an alpha (α) value, a shape parameter, of 1.47 was used. For the beta (β) value, a scale parameter, 30.17 was used for males and 32.8 for females. The α—and β values were chosen based on the best fit with the observed data provided by SimCYP^®^. The age–height and weight–height relations were overpredicted for the *Black South African population*, starting from the *Sim-NEur Caucasian population* [35]. New values were inserted by SimCYP^®^, as shown in Table 2.

The body surface area (BSA) estimates using the DuBois and DuBois equation are lower than the observed values for Black South African subjects [36]. Therefore the Nwoye formula (Equation (1)) was applied based on coating and planimetry measurements on a male, 18- to 55-year-old, non-obese, Nigerian population of 20 subjects [37].
(1)$$\mathrm{BSA}\;\left(m^{2}\right)={\mathrm{weight}}^{-0.2620}\times 0.001315\times {\;\mathrm{height}}^{1.2139}$$

New equations were implemented by SimCYP^®^ to estimate the glomerular filtration rate (GFR) in the *Black South African population* for males with serum creatinine (Scr) levels lower than 80 µmol/L (Equation (2)) and higher than 80 µmol/L (Equation (3)) and females with serum creatinine (Scr) levels lower than 62 µmol/L (Equation (4)) and higher than 62 µmol/L (Equation (5)). For Black males, a factor of 163 was implemented instead of 141 for White males; for Black women, a factor of 166 was implemented instead of 144 for White women [38].
(2)$$\mathrm{GFR}=163\times {\left(\frac{\mathrm{Scr}}{80}\right)}^{-0.411}\times 0.993^{\mathrm{Age}}$$
(3)$$\mathrm{GFR}=163\times {\left(\frac{\mathrm{Scr}}{80}\right)}^{-1.209}\times 0.993^{\mathrm{Age}}$$
(4)$$\mathrm{GFR}=166\times {\left(\frac{\mathrm{Scr}}{62}\right)}^{-0.329}\times 0.993^{\mathrm{Age}}$$
(5)$$\mathrm{GFR}=166\times {\left(\frac{\mathrm{Scr}}{62}\right)}^{-1.209}\times 0.993^{\mathrm{Age}}$$

A Lua Code was used for scripting in the user-defined GFR function in SimCYP^®^. It is important to note that AFB1 is not renally cleared, so the GFR will not impact the PK of AFB1.

CYP abundances and phenotype frequency for eight CYP450 enzymes and the selected parameters are listed for the *Black South African* population, *Sim*-*NEur Caucasian* population and the *Sim-Chinese healthy volunteer* population (Table 3) [39].

Since AFB1 is classified as Group 1 by IARC as carcinogenic to humans, it cannot be administered to humans during in vivo trials [2]. However, there is one available in vivo trial conducted on four healthy Chinese volunteers (age range 32–65 y/o, 25% female) following 30 ng of ^14^C-labelled AFB1 orally in a fasted state; the pharmacokinetic parameters of AFB1 are reported by Jubert et al. (2009) [28]. The PBPK model developed for AFB1 from this study (as mentioned in the previous section) was used in the SimCYP^®^ *Sim-Chinese healthy volunteer* population to predict the plasma concentration (C_p_) versus time profile, as presented in the work of Jubert et al. (2009), allowing a comparison between the predicted and observed C_p_-time curves [28]. For the PBPK model trials, a population size of 100 was selected, consisting of 10 trials with 10 subjects per trial. For the whole population, a fasted prandial state was selected. An age range of 32–65 y and a proportion of females of 25% were selected. A single dose of 30 ng of AFB1 was used, as in the in vivo trial by Jubert et al. (2009) [28]. The predictions were to be considered successful and the model considered verified if the predicted PK parameters fell within a twofold range of the in vivo values [40]. The average fold error (AFE) and the absolute average fold error (AAFE) were calculated for all plasma concentrations over time, based on Equations (6) and (7), respectively, to evaluate model performance on plasma concentration predictions of AFB1, as simulated in the healthy Chinese volunteers [41].
(6)$$\mathrm{AFE}=10^{\left[\frac{\sum \log10\left(\frac{\mathrm{predicted}}{\mathrm{observed}}\right)}{\mathrm{number}\;\mathrm{of}\;\mathrm{observations}}\right]\;}$$
(7)$$\mathrm{AAFE}=10^{\left[\frac{\sum \mathrm{ABS}\left(\log10\left(\frac{\mathrm{predicted}}{\mathrm{observed}}\right)\right)}{\mathrm{number}\;\mathrm{of}\;\mathrm{observations}}\right]\;}$$

Since several enzyme abundance and physiological parameters in Black South African, Caucasian and Chinese populations were different, which may affect the AFB1 disposition in those populations, an inter-ethnic PK comparison was performed by simulating the PK of AFB1 in the *Black South African* population, *Sim-NEur Caucasian* population and *Sim-Chinese healthy volunteer* population, following a 30 ng AFB1 single dose in a fasted state. For each population, a trial was performed with 5000 subjects. The proportion of females was 50%, and the age range within each virtual population was between 20 and 50 years old.

An insight into the currently used drugs in South Africa was provided by the South African Essential Medicine List of the World Health Organization (WHO), containing 192 drugs [42]. A selection was made based on the CYP450 enzymes (CYP3A4/CYP1A2 involvement). Drugs that are a CYP3A4 and/or CYP1A2 substrate, inhibitor or inducer in the SimCYP^®^ library were used in the simulations since AFB1 is metabolised by those CYP450 enzymes. Drugs that are clinically administered in low doses were chosen since AFB1 exposure is considered low dose, and, therefore, the possible impact of AFB1 on drugs could be observed. Table 4 shows the list of drugs used in the interaction simulations in the *Black South African* population when AFB1 is a substrate and an inhibitor. A trial design of 30 days was chosen, with a daily intake of both the drug (common administered dose) and AFB1 (30 ng). In the virtual population, 10 trials of 10 subjects were performed, with a proportion of females of 50% and with an age range of 20–50 years. The consumption of 30 ng AFB1 was based on the in vivo trial by Jubert et al. (2009), where 30 ng equals one-twentieth of the US maximum limit of 20 µg/kg AFs in a 30 g peanut butter sandwich [43]. In the EU, the maximum limit for AFs in peanut products is 15 µg/kg [44].

## 3. Results

The model was evaluated by plotting its predictions with the available plasma concentration versus the time profiles of the four healthy volunteers, as shown in Figure 3. Figure 3 represents the plasma concentrations of AFB1 versus time, as predicted by the PBPK model, and the observed data points in the four subjects from the in vivo trial [28].

The observed data from Jubert et al. (2009) and from the predictions using the PBPK model, i.e., C_max_, AUC_0–24 h_, and T_max_ are listed in Table 5.

The simulated profile of AFB1 was comparable to the clinical data, and the predicted mean C_max_, AUC_0–24 h_, and T_max_ for AFB1 administrated to healthy Chinese volunteers were within 1.08-fold, 0.80-fold and 1.61-fold of the observed values, respectively.

Figure 4 shows the simulated AFB1 C_p_ versus time profile in the *Sim*-*Chinese healthy volunteer* population, the *Sim-NEur Caucasian* population and the *Black South African* population (5000 subjects per population) exposed to a single dose of 30 ng AFB1. The mean C_max_ (pg/mL), T_max_ (h), AUC (pg/mL.h) and CL (L/h) are presented in Table 6.

As observed in Figure 4 and Table 6, outcomes can be different in populations exposed to the same compound and dose. Clearance in the *Black South African* population is 1.90-fold higher than the clearance in the *Sim-Chinese healthy volunteer* population. The clearance values clearly show that different populations demonstrate differences in drug disposition.

Table 7 shows the predicted PK parameters of AFB1, following a once-daily dose of 30 ng in the *Black South African* (10 trials of 10 subjects) population, with or without commonly used drugs administration (daily single dose), over a time span of 30 days. Data of the combinations (Table 4), where no difference in PK parameters was observed between AFB1 alone and in combination, are not included in Table 7.

## 4. Discussion

A PBPK model can be developed for every compound if enough high-quality data and population information are available. In clinical phases in the development of a new drug, PBPK modelling is often applied for a variety of purposes. It follows the 3R principle, and it can be used to predict PK parameters by simulating trials in, e.g., human populations that were not performed in real life. In vitro data on metabolism are required to unravel the metabolic pathways and their importance next to enzyme involvement. In vitro–in vivo extrapolation (IVIVE) is, therefore, the ultimate tool: by performing in vitro experiments on human cells or microsomes, one can determine the metabolic parameters needed to build a PBPK model. In this case, not much data were available for AFB1; therefore, in vitro experiments were performed and certain parameters were predicted based on the physicochemical characteristics of AFB1 [31]. The model was built using a bottom-up approach without any fitting. IVIVE-PBPK is a very suitable approach to check for potential interactions between compounds to which people are chronically exposed (such as drugs, food contaminants etc.). Since mycotoxins, among other food contaminants, are a part of the human diet, interactions need to be considered. Because of the low concentration, in chronic exposure, mycotoxins will potentially not have an effect on the PK of drugs, but drugs might have an impact on the PK of mycotoxins, leading to higher exposure to the parent compound or higher exposure to metabolites, potentially leading to increased or decreased toxicity. For compounds that cannot be experimentally tested on humans, physiologically based mechanistic software tools such as PBPK modelling are the go-to method. Since exposure to mycotoxins is largely unknown, instructions might be given in case certain drugs are administered, and mycotoxin exposure is expected based on the geographic region of the world (e.g., the African continent). Examples of what to mention on the patient information leaflet of concerned drugs might be symptoms of aflatoxicosis, or special attention might be drawn to mycotoxin awareness to avoid homegrown, mouldy foods.

A PBPK model is verified if the predicted data are within twofold of the observed data. Since the developed PBPK model seems to predict the PK parameters within 0.80–1.61-fold (Table 5) of the scarce available in vivo data, the model can be considered verified as a PBPK model. Of note is that the used MRP3 value from the literature comes from AFB1 metabolites and is used as a surrogate for AFB1. The results of the simulations run in three different populations (Table 6), i.e., the *Sim-Chinese healthy volunteer* population, the *NEur Caucasian* population and the *Black South African* population, indicate that it is necessary and important to consider different populations. The PK parameters C_max_, T_max_ and AUC_0–24 h_ are slightly different between the different populations (fold differences of 1.28, 1.17 and 1.6 between the Black South African and Chinese healthy volunteer populations), but it is clear that different populations have an impact on AFB1 CL, with a 1.90-fold CL difference between the Black South African and Chinese healthy volunteer population. Different regions in the world have different populations with a variety of genotypes and phenotypes. The disposition of drugs can, therefore, be altered in different populations and needs to be accounted for. It is relevant to further elaborate on this since there is still a lack of granularity in how populations are subdivided, e.g., people from the African continent are considered as one population, whereas in reality, different regions have differing populations, leading to differences in the PK of compounds. A pilot Black South African population has been built by SimCYP^®^ using the scarce data that are available and using information from other countries rather than South Africa alone (i.e., Nigeria). More data on the different population components (i.e., liver, kidney, skin, GI tract, tissue composition, tissue flow rates, tissue pore size, brain, lung, etc.) for the *Black South African* population dataset should be retrieved to optimise this pilot population.

As mentioned in the Materials and Methods section, concomitant intake of AFB1 with a drug (CYP1A2/3A4 perpetrators and substrates) was simulated using SimCYP^®^ (version 21) and is shown in Table 7. The cases where AFB1 is selected as an inhibitor do not impact the PK parameters of the tested drugs. AFB1 exposure is much lower compared to chronic drug exposure, often daily dosed in milligrams. Thus, it can be stated that AFB1 will not have a major impact on the PK of drugs at concentrations occurring in chronic exposure, not even at high acute exposure concentrations. However, drugs can have an impact on the PK of AFB1 and, potentially, other mycotoxins. Since the interaction between AFB1 and certain drugs (carbamazepine, efavirenz, phenobarbital, phenytoin and rifampicin) can lead to an increased clearance of AFB1 and lower AUC, up to 4.13 CL fold-difference and 0.26 AUC fold-difference with AFB1 alone, it indicates that more metabolites are formed. In contrast, interactions between AFB1 and other drugs (atazanavir, ciprofloxacin, fluconazole and ritonavir) lead to a lower clearance of AFB1 in the human body, consequently having a higher AUC, up to 0.47 and 2.5 fold-difference with the intake of AFB1 alone, where metabolites are less extensively formed. Carbamazepine induces both CYP1A2 and CYP3A4 and is itself a CYP3A4 substrate. Efavirenz is next to a CYP2B6 inducer/substrate, and the CYP2C19 inducer is also a CYP3A4 inducer. Phenobarbital, phenytoin and rifampicin are CYP1A2 and CYP3A4 inducers. Since CYP1A2 and CYP3A4 are responsible for the formation of carcinogenic AFBO metabolites, it might be possible that people taking one of these drugs are more exposed to carcinogenic metabolites of AFB1. In the case of rifampicin, there is less danger since it is an antibiotic, mostly for acute use, while carbamazepine and phenytoin are often chronically used and might cause higher exposure to AFBO in the case of mycotoxin-contaminated diets. When AFB1 is co-administrated with CYP3A4 and/or CYP1A2 inhibitors, there is a lower risk of being exposed to the AFBO metabolites of AFB1 in the case of a mycotoxin-contaminated diet.

## 5. Conclusions

It has been shown that CYP3A4/CYP1A2 inducer or inhibitor drugs have an impact on the PK of AFB1, potentially leading to higher or lower exposure to carcinogenic metabolites (AFBOs). With regard to the impact on risk assessment, there is the margin of exposure (MOE) for genotoxic and carcinogenic compounds, which is the ratio of the exposure level at which human health is not harmed and the estimated level of human exposure [45]. Consequently, it might turn out that the MOE needs to be reconsidered in the case of co-exposure to certain drugs regarding the potential effect on the disposition of mycotoxins. Furthermore, drugs that impact the PK of AFB1 should have the symptoms of aflatoxicosis in the patient information leaflet, broadening the awareness on this topic and making physicians and pharmacists aware of possible interactions between food and drugs. In conclusion, IVIVE-PBPK is a valuable tool, especially in cases where no human in vivo trials can be performed. It can be stated that mycotoxins will not likely have an impact on the PK of drugs regarding the differences in exposure/dosage. Drugs, on the other hand, might have an impact on the metabolism of mycotoxins, potentially leading to more or less toxicity, depending on the type of interaction (inhibition/induction) with enzymes and/or transporters and depending on detoxification or bioactivation pathways.

## Figures and Tables

**Figure 1 pharmaceutics-15-00894-f001:**
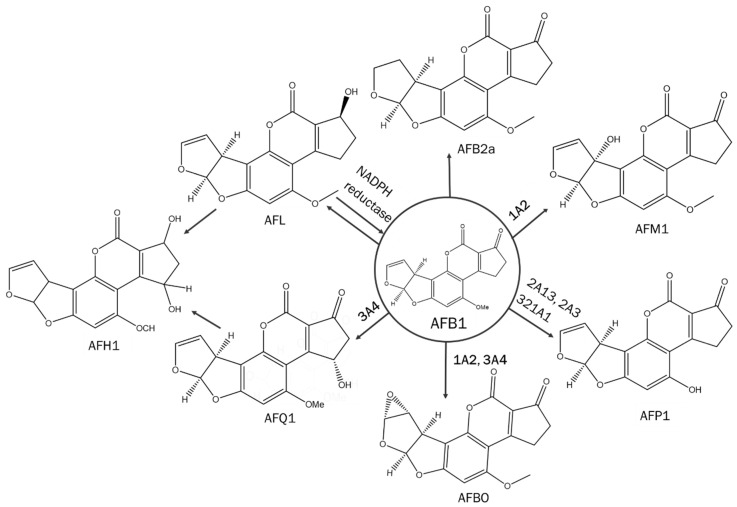
Schematic representation of the hepatic metabolic pathway of aflatoxin B1 (AFB1) in humans. AFB1 = aflatoxin B1; AFB2a = aflatoxin B2a; AFBO = aflatoxin-endo/exo-8,9-epoxide; AFH1 = aflatoxin H1; AFL = aflatoxicol; AFM1 = aflatoxin M1; AFP1 = aflatoxin P1; AFQ1 = aflatoxin Q1; NADPH reductase = nicotinamide-adenine-dinucleotide-phosphate reductase.

**Figure 2 pharmaceutics-15-00894-f002:**
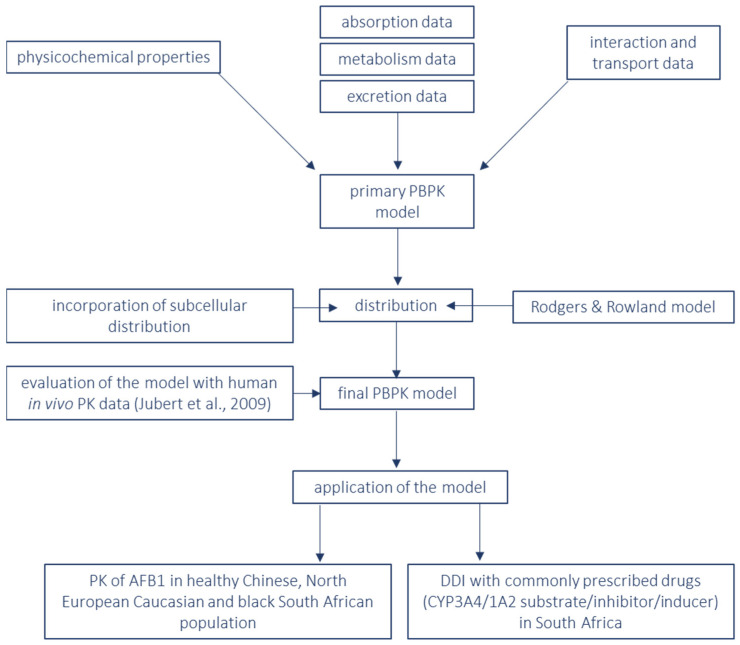
Flowchart of the bottom-up approach for building a PBPK model for substrate aflatoxin B1 and its applications [28]. PBPK = physiologically based pharmacokinetics; PK = pharmacokinetics; AFB1 = aflatoxin B1; DDI = drug–drug interaction; CYP = cytochrome P 450.

**Figure 3 pharmaceutics-15-00894-f003:**
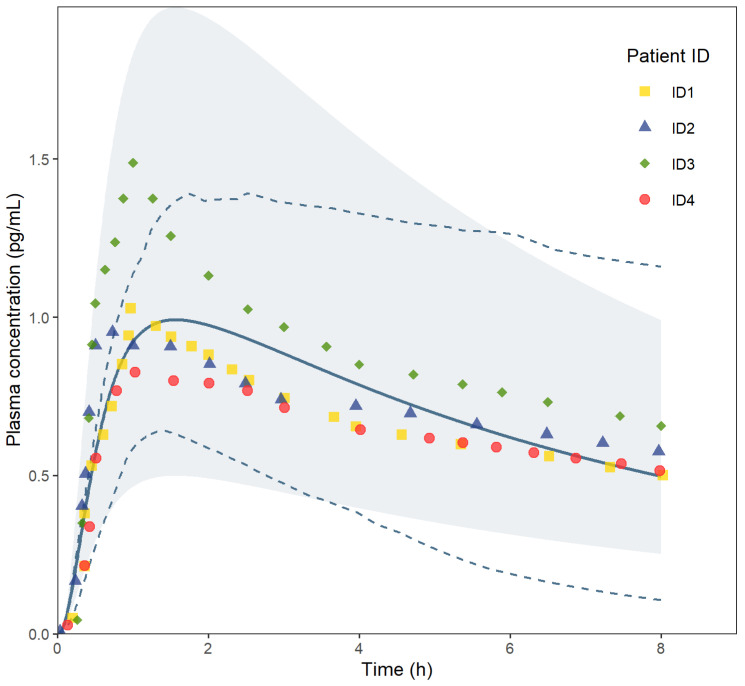
AFB1 plasma concentration (pg/mL) versus time (h) curve: predicted data by the PBPK model are shown by the blue line, and the observed data points of the four subjects from the in vivo trial are represented as coloured symbols (each colour represents a subject). The light blue area shows the twofold prediction. The dashed blue lines give the 5th–95th percentiles of the predicted plasma concentrations.

**Figure 4 pharmaceutics-15-00894-f004:**
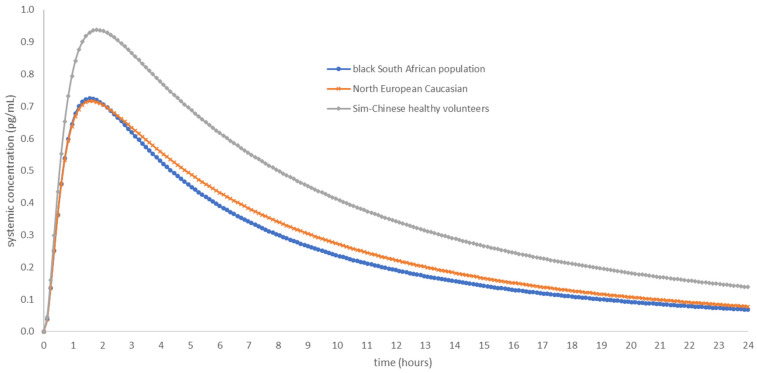
Systemic AFB1 concentration (pg/mL) versus time (h) curve in a *Sim-Chinese healthy volunteer* population of 5000 subjects (grey line), in a *Sim-NEur Caucasian* population of 5000 subjects (orange line) and in a *Black South African* population of 5000 subjects (blue line) over a time span of 24 h.

**Table 1 pharmaceutics-15-00894-t001:** Physicochemical properties, blood-binding, absorption, distribution, elimination, interaction and transport information for AFB1.

Physicochemical Properties and Blood Binding		
Parameter	Model Input Value	Reference
M_w_ (g/moL)	312.27	[29]
LogP_o:w_	1.60	[29]
Compound type	neutral	[29]
ECCS	Class 2	[30]
B:P	1.03	Predicted *
f_u,p_	0.17	[26]
Absorption		
f_a_	0.99	Predicted *
k_a_	2.39	Predicted *
P_eff,man_ (10^−4^ cm/s)	5.47	predicted *
P_trans,0_ (10^−6^ cm/s)	135.8	Predicted *
Absorption and Metabolism (ADAM) Model		
Formulation—Diffusion Layer Model—Aqueous Phase Solubility—Solid State 1		
S_0_	1.24	Predicted *
Distribution		
Full PBPK model		
V_ss_ (L/kg)	0.33	prediction method 3
Tissue: plasma partition coefficients/K_p_ scalar = 1		
Adipose	0.44	predicted *
Bone	0.15	predicted *
Brain	0.55	predicted *
Gut	0.36	predicted *
Pancreas	0.26	predicted *
Heart	0.37	predicted *
Kidney	0.36	predicted *
Liver	0.44	predicted *
Lung	0.33	predicted *
Muscle	0.23	predicted *
Skin	0.28	predicted *
Spleen	0.44	predicted *
Elimination		
Enzyme kinetics		
CYPs	Recombinant	
CYP3A4 K_m_	49.60 µM	Experimental [31]
CYP3A4 V_max_	88.10 pmol/min/pmol CYP	Experimental [31]
CYP3A4 ISEF	0.50	Experimental [31]
CYP1A2 K_m_	58.20 µM	Experimental [31]
CYP1A2 V_max_	199.00 pmol/min/pmol CYP	Experimental [31]
ISEF	1.42	Experimental [31]
Interaction		
CYP1A2 K_i_	10.2 µM	Experimental [31]
Transport		
Using permeability limited liver model		
CL_PD_ (mL/min/million hepatocytes)	0.05	derived from [32]
f_u,IW_	0.35	predicted *
f_u,EW_	0.17	predicted *
Drug concentration for passive permeability: unbound (ionized and unionized) Sinusoidal: Efflux: ABCC3 (MRP3)		
J_max_ (pmol/min/million cells)	180.00	[33]
K_m_ (µM)	0.19	
f_u,inc_	1.00	
RAF/REF	2.50	[34]

* Predicted using SimCYP. M_w_ = molecular weight; logP_o:w_ = logarithm of the octanol–water partition coefficient; ECCS = Extended Clearance Classification System; B:P = blood to plasma ratio; f_u,p_ = fraction unbound in plasma; f_a_ = fraction absorbed, available from dosage form; k_a_ = first-order absorption rate constant; P_eff,man_ = human jejunum permeability; P_trans,0_ = intrinsic transcellular permeability; S_0_ = solubility; PBPK = physiologically based pharmacokinetics; V_ss_ = volume of distribution at steady-state; K_p_ = tissue:plasma partition coefficient; CYPs: cytochrome P450 enzymes; K_m_ = Michaelis–Menten constant; V_max_ = maximum reaction velocity; ISEF = intersystem extrapolation factor; K_i_ = inhibitor constant; CL_PD_ = passive diffusion clearance; f_u,IW_ = fraction unbound in intracellular water; f_u,EW_ = fraction unbound in extracellular water; MRP3 = multidrug resistance-associated protein 3; J_max_ = in vitro maximum rate of transporter-mediated efflux; f_u,inc_ = fraction unbound in the incubation; RAF/REF = relative activity factor/relative expression factor.

**Table 2 pharmaceutics-15-00894-t002:** Height–weight parameters for the *Black South African* population for both males and females.

**Height**			
**MALE**		**FEMALE**	
C0	161.780	C0	155.376
C1	0.359	C1	0.207
C2	−0.00429	C2	−0.00268
CV (%)	7.33	CV (%)	5.83
**Weight**			
**MALE**		**FEMALE**	
C0	2.97	C0	3.19
C1	0.007	C1	0.007
CV (%)	21.1	CV (%)	26.38

C0, C1 and C2 = coefficients for the dependence of body weight on height (no units); CV (%) = coefficient of variation for the dependence of weight on height in percentage.

**Table 3 pharmaceutics-15-00894-t003:** A summary of CYP450 abundances and phenotype frequency for eight CYP450 enzymes and of selected parameters in the *Black South African* population, *Sim-NEur Caucasian* population and *Sim-Chinese healthy volunteer* population.

*Black South African*				*Sim-NEur Caucasian*			*Sim-Chinese Healthy Volunteer*		
CYP450 Enzyme	Abundance (pmol/mg Protein)/CV	Phenotype Frequency		Abundance (pmol/mg Protein)/CV	Phenotype Frequency		Abundance (pmol/mg Protein)/CV	Phenotype Frequency	
		EM	PM		EM	PM		EM	PM
CYP1A2	52/67%	1	0	52/67%	1	0	42/50%	1	0
CYP2B6	6.9/122%	0.85	0.15	21.6/68%	0.40	0.10	6.7/63%	0.52	0.07
CYP2C9	73/ 54%	0.98	0.02	77.7/64%	0.66	0.019	87.6/55%	0.93	0.003
CYP2C19	14/106%	0.96	0.04	4.4/52%	0.42	0.023	4.4/52%	0.40	0.13
CYP2D6	8/61%	0.97	0.03	9.4/65%	0.57	0.08	10.47/65%	0.60	0.003
CYP3A4	137/41%	1	0	137/41%	1	0	120/33%	1	0
CYP3A5	71/78%	0.82	0.18	103/65%	0.17	0.83	82.3/68%	0.42	0.58
CYP3A7	35.4/61%	0.12	0.88	35.4/61%	0.12	0.88	14/71%	0.12	0.88
parameter	parameter value	CV (%)		parameter value	CV (%)		parameter value	CV (%)	
LV (L)	1.924	12		1.651	12		1.403	12	
MPPGL (mg/g)	39.79	N.A.		39.79	N.A.		39.45	N.A.	
LD (g/L)	1080	N.A.		1080	N.A.		1080	N.A.	
Hematocrit (%) (male)	43	6.51		43	6.5		45.3	9.5	
Hematocrit (%) (female)	38	7.13		38	7.1		40.5	10.9	
AGP (g/L) (male)	0.811	15		0.739	23		0.683	23	
AGP (g/L) (female)	0.791	13		0.715	24		0.575	24	
HSA (g/L) (male)	50.34	10		50.34	10		50.34	10	
HSA (g/L) (female)	49.38	10		49.38	10		49.38	10	
Weibull α (male)	1.47	N.A.		5.47	N.A.		1.5	N.A.	
Weibull β (male)	30.17	N.A.		66.5	N.A.		19	N.A.	
Weibull α (female)	1.47	N.A.		5.22	N.A.		4.48	N.A.	
Weibull β (female)	32.8	N.A.		68.57	N.A.		53.4	N.A.	

CYP450 = cytochrome P 450; CV (%) = coefficient of variation; EM = extensive metaboliser; PM = poor metaboliser; LV = liver volume; MPPGL = microsomal protein per gram liver; N.A. = not applicable; LD = liver density; AGP = α_1_-acid glycoprotein; HSA = human serum albumin.

**Table 4 pharmaceutics-15-00894-t004:** Selection of the South African Essential Medicines list from WHO, used in SimCYP^®^ simulations with AFB1, the drug administration dose (mg), the CYP3A4/CYP1A2 substrate/inhibitor or inducer and the drug class.

Drug	Dose (mg) QD	CYP3A4/CYP1A2 Substrate/Inhibitor/Inducer	Drug Class
artemether	20	CYP3A4 substrate	antimalarial
atazanavir	200	CYP3A4 substrate CYP3A4 inhibitor	protease inhibitor
carbamazepine	200	CYP3A4 substrate CYP3A4 inducer	anticonvulsant
ciprofloxacin	250	CYP1A2 inhibitor	quinolone antibiotics
efavirenz	600	CYP3A4 inducer	non-nucleoside reverse transcriptase inhibitor
ethinylestradiol	0.035	CYP3A4 substrate CYP1A2 inhibitor	estrogen
phenobarbital	100	CYP3A4 and CYP1A2 inducer	barbiturate
phenytoin	100	CYP3A4 and CYP1A2 inducer	anticonvulsant
fluconazole	50	CYP3A4 inhibitor	triazole antifungal
fluoxetine	20	CYP3A4 inhibitor	selective serotonin reuptake inhibitor
midazolam	5	CYP3A4 substrate	benzodiazepine
nifedipine	20	CYP3A4 substrate	calcium channel blocker
rifampicin	600	CYP1A2 inducer CYP3A4 inducer	antimycobacterial
ritonavir	600 BID	CYP3A4 substrate CYP3A4 inhibitor	protease inhibitor
simvastatin	20	CYP3A4 substrate	statins

QD = quaque die; CYP = cytochrome P 450 enzymes; BID = bis in die.

**Table 5 pharmaceutics-15-00894-t005:** Overview of the mean observed data from Jubert et al. (2009) [28] and the simulated data in the *Sim-Chinese healthy volunteer* population. The prediction/observed ratio is given in the right column.

	Observed Data (Mean ± SD)	Predicted Data (Mean ± SD)	Predicted/Observed Ratio
C_max_ (pg/mL)	0.941 ± 0.154	1.02 ± 0.035	1.08
AUC_0–24 h_ (pg/mL.h)	12.4 ± 1.8	9.87 ± 0.825	0.80
T_max_ (h)	1.02 ± 0.31 h	1.64 ± 0.075 h	1.61
AFE on C_p_		1.12	
AAFE on C_p_		1.35	

C_max_ = maximum plasma concentration; AUC_0–24 h_ = area under the curve from 0–24 h; T_max_ = time at which C_max_ is achieved; AFE = average fold error; C_p_ = plasma concentration; AAFE = absolute average fold error.

**Table 6 pharmaceutics-15-00894-t006:** Summary of the predicted PK parameters for healthy Chinese volunteers, North European Caucasians and Black South Africans by simulating 5000 virtual subjects.

	Sim-Chinese Healthy Volunteers	North European Caucasian	Black South African
mean C_max_ (pg/mL)	0.967	0.740	0.755
mean T_max_ (h)	1.92	1.67	1.64
mean AUC_0–24 h_ (pg/mL.h)	9.85	6.78	6.24
mean CL (L/h)	4.62	6.52	8.78

C_max_ = maximum plasma concentration; T_max_ = time at which C_max_ is achieved; AUC_0–24 h_ = area under the curve from 0–24 h; CL = clearance.

**Table 7 pharmaceutics-15-00894-t007:** Overview of C_max_, T_max_, AUC, C_min_ and CL (D/AUC) (+mean error) for AFB1 alone and in combination with other drugs.

	AFB1 Alone		AFB1 + Drug		Ratio of PK Parameters (with Drug/without Drug)
		ME	+Atazanavir (200 mg) QD	ME	
C_max_ (pg/mL)	1.19	0.076	1.69	0.12	1.39
T_max_ (h)	0.96	0.055	1.09	0.07	1.14
AUC_0-inf_ (pg/mL.h)	11.7	1.21	23.7	2.54	2.09
C_min_ (pg/mL)	0.18	0.0091	0.55	0.0055	3.10
CL (L/h)	5.82	0.58	3.12	0.31	0.54
			+carbamazepine (200 mg) QD		
C_max_ (pg/mL)	1.21	0.08	1.00	0.063	0.83
T_max_ (h)	0.96	0.06	0.92	0.055	0.96
AUC_0-inf_ (pg/mL.h)	12.09	1.24	8.92	0.91	0.74
C_min_ (pg/mL)	0.19	0.00096	0.12	0.00048	0.62
CL (L/h)	5.69	0.57	7.28	0.72	1.28
			+ciprofloxacin (250 mg) QD		
C_max_ (pg/mL)	1.20	0.075	1.52	0.093	1.27
T_max_ (h)	0.95	0.055	1.28	0.08	1.35
AUC_0-inf_ (pg/mL.h)	11.8	1.22	17.42	1.68	1.47
C_min_ (pg/mL)	0.19	0.00092	0.28	0.00188	1.51
CL (L/h)	5.86	0.58	3.21	0.31	0.55
			+efavirenz (600 mg) QD		
C_max_ (pg/mL)	1.21	0.082	0.80	0.0053	0.66
T_max_ (h)	0.95	0.06	0.71	0.045	0.75
AUC_0-inf_ (pg/mL.h)	12.20	1.25	4.41	0.44	0.36
C_min_ (pg/mL)	0.20	0.0090	0.03	0.00006	0.16
CL (L/h)	5.91	0.59	15.52	1.62	2.63
			+phenobarbital (100 mg) QD		
C_max_ (pg/mL)	1.21	0.078	0.78	0.055	0.64
T_max_ (h)	0.95	0.06	0.80	0.05	0.84
AUC_0-inf_ (pg/mL.h)	12.1	1.25	5.39	0.56	0.45
C_min_ (pg/mL)	0.20	0.009	0.05	0.00047	0.25
CL (L/h)	5.91	0.59	13.60	1.38	2.30
			+phenytoin (100 mg) QD		
C_max_ (pg/mL)	1.21	0.08	0.95	0.063	0.79
T_max_ (h)	0.96	0.06	0.89	0.055	0.93
AUC_0-inf_ (pg/mL.h)	12.2	1.26	8.09	0.82	0.66
C_min_ (pg/mL)	0.20	0.0095	0.11	0.00284	0.55
CL (L/h)	5.78	0.57	8.78	0.87	1.52
			+fluconazole (50 mg) QD		
C_max_ (pg/mL)	1.20	0.064	1.35	0.07	1.13
T_max_ (h)	0.95	0.06	1.00	0.06	1.05
AUC_0-inf_ (pg/mL.h)	12.0	0.8	15.0	0.95	1.25
C_min_ (pg/mL)	0.19	0.00113	0.267	0.00201	1.41
CL (L/h)	5.88	0.92	4.75	0.79	0.81
			+rifampicin (600 mg) QD		
C_max_ (pg/mL)	1.19	0.08	0.64	0.051	0.54
T_max_ (h)	0.96	0.055	0.71	0.045	0.74
AUC_0-inf_ (pg/mL.h)	11.7	1.21	3.02	0.31	0.26
C_min_ (pg/mL)	0.18	0.0090	0.013	0.00001	0.07
CL (L/h)	5.82	0.575	24.03	2.47	4.13
			+ ritonavir (600 mg) BID		
C_max_ (pg/mL)	1.21	0.078	1.95	0.15	1.56
T_max_ (h)	0.95	0.055	1.10	0.07	1.16
AUC_0-inf_ (pg/mL.h)	12.2	1.21	29.0	3.1	2.50
C_min_ (pg/mL)	0.20	0.0090	0.75	0.078	3.75
CL (L/h)	5.91	0.575	2.78	0.27	0.47

ME = mean error; C_max_ = maximal plasma concentration of the mean concentration time profile; T_max_ = time at which the maximal plasma concentration occurs in the mean concentration time profile; AUC_0-inf_ = area under the concentration curve from zero to infinity; C_min_ = minimal plasma concentration of the mean concentration time profile; CL = clearance; QD = quaque die; BID = bis in die.

## Data Availability

The data presented in this study are available on request from the corresponding author. The data are not publicly available due to their use in further research.

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
