# Peer review of "Building a Human Physiologically Based Pharmacokinetic Model for Aflatoxin B1 to Simulate Interactions with Drugs"

_pharmaceutics, 2023, doi:10.3390/pharmaceutics15030894_

Round 1

Reviewer 1 Report

Dear Authors,

Thank you for given opportunity to review this manuscript. Your research deals with the interesting topic of the impact of  aflatoxin and its metabolites on human health, in patient undergoing pharmacotherapy. That was good written article with relevant data and conclusions. 

My main question is: How to apply this findings in everyday clinical practice? What is a clinical meaning of your results? This could be added to conclusion.

I suggest a continuation of research aimed with clear guidelines for physicians and pharmacists.

Sincerely,

Reviewer 2 Report

-The method of data validation is not explained in the manuscript. 

-The results from the tables were not discussed very well, as for the reader, it shows that this is more explanation regarding the software simulation capabilities.

Reviewer 3 Report

The study is relevant and of general interest to the readers of this journal, containing interesting findings relating to aflatoxin B1 and simulated interactions involving the P450 isoforms.

Concerning the main factor of novelty, we can mention the PBPK simulation where is shown that drugs targeting CYP 3A4/1A2 interfere with the PK of aflatoxin B1 and the level of exposure to carcinogenic metabolites.

The introduction presents a well-structured format, perfectly integrating the main issues under study while citing recent bibliography.

We found this article well written, with a good and clear organization of the contents and adequate methodology for drug kinetic modeling and model validation. Only minor comments will be done focusing on the improvement of this subject.

The cited references are recent and appropriate to the discussion.

The availability of data by the authors reflects the clarity and openness desirable in science. We congratulate the authors on this.

Minor comments:

#1_L231-235_Table 5_ The "Mean+/- SD" label (to indicate the uncertainty around the estimated mean) is lacking. Please add the appropriate label.

#2_The Conclusions item is lacking. Please include the Conclusions item.

#3_ L258-268_Table 7_ Please include the precision of the predicted PK values (SD, CV).

#4_L353-354_ A “error 404_file not found” appears when we attempt to get the link. Please check this.
